# Robust Motor Imagery Tasks Classification Approach Using Bayesian Neural Network

**DOI:** 10.3390/s23020703

**Published:** 2023-01-08

**Authors:** Daily Milanés-Hermosilla, Rafael Trujillo-Codorniú, Saddid Lamar-Carbonell, Roberto Sagaró-Zamora, Jorge Jadid Tamayo-Pacheco, John Jairo Villarejo-Mayor, Denis Delisle-Rodriguez

**Affiliations:** 1Department of Automatic Engineering, University of Oriente, Santiago de Cuba 90500, Cuba; 2Electronics, Communications and Computing Services Company for the Nickel Industry, Holguín 80100, Cuba; 3Independent Researcher, 33720 Tampere, Finland; 4Department of Mechanical Engineering, University of Oriente, Santiago de Cuba 90500, Cuba; 5Department of Electrical and Electronic Engineering, Federal University of Santa Catarina, Florianopolis 88040-900, SC, Brazil; 6Postgraduate Program in Neuroengineering, Edmond and Lily Safra International Institute of Neurosciences, Santos Dumont Institute, Macaiba 59280-000, RN, Brazil

**Keywords:** brain–computer interface, Bayesian Neural Network, variational inference, uncertainty estimation, classification with reject option

## Abstract

The development of Brain–Computer Interfaces based on Motor Imagery (MI) tasks is a relevant research topic worldwide. The design of accurate and reliable BCI systems remains a challenge, mainly in terms of increasing performance and usability. Classifiers based on Bayesian Neural Networks are proposed in this work by using the variational inference, aiming to analyze the uncertainty during the MI prediction. An adaptive threshold scheme is proposed here for MI classification with a reject option, and its performance on both datasets 2a and 2b from BCI Competition IV is compared with other approaches based on thresholds. The results using subject-specific and non-subject-specific training strategies are encouraging. From the uncertainty analysis, considerations for reducing computational cost are proposed for future work.

## 1. Introduction

A Brain–Computer Interface (BCI) provides a direct communication pathway between the user’s brain and external devices/software in order to monitor and/or repair sensorimotor and cognitive functions [1,2,3]. The BCI’s input receives brain commands typically via electroencephalograms (EEG), which is a non-invasive, low-cost, high temporal resolution, and portable technique. Many EEG-based BCIs have been developed by using different paradigms, such as P300 event-related potentials [4,5], steady-state visual evoked potentials (SSVEP) [6,7], and sensorimotor rhythms (SMR) [8,9,10]. Systems using SMR have demonstrated promising results and great potential for training and recovering motor skills through various types of motor imagery tasks. However, the poor EEG signal-to-noise ratio and intra-and inter-subject variability make the MI classification still a challenge.

Deep learning techniques for MI classification have shown better performance than handcrafted feature extraction methods with respect to accuracy [11,12,13,14,15,16,17,18,19]. This success of deep learning also appears in other research areas, such as computer vision, image, and language processing, among others. Nevertheless, these approaches are deterministic architectures, and hence they lack strategies to measure classification trust.

The source of uncertainty in deep learning models may come from data input because of noise and inaccurate measurement (aleatoric uncertainty). Uncertainty from other sources due to lack of knowledge is termed epistemic uncertainty [20,21]. For this reason, uncertainty analysis is crucial to providing robust and reliable classification, which is essential in developments that require low error tolerance, such as medical and autonomous navigation systems. For instance, in these systems, classification mistakes may produce poor decisions, and consequently fatal events. In medical applications, uncertainty analysis has been widely used for disease diagnoses, such as COVID-19 [22], tuberculosis [23], ataxia [24], cancer [25], diabetic retinopathy [26,27], and epileptogenic brain malformations [28]. To the best of our knowledge, strategies for uncertainty analysis have been little explored in MI-based BCIs, excluding a previous study [29] that applied Monte Carlo dropout techniques with this purpose, considered as an approximate Bayesian inference in deep Gaussian processes [21]. The current paper follows a different method by applying BNN.

During classification, the output score vector produced by deterministic architectures is often erroneously interpreted as model confidence (as it appears as probabilities due to the Softmax activation function). Various studies have demonstrated that, even when the highest score is large, the score vector can be a poor predictor of uncertainty. For instance, in [30], the authors perturbed a single pixel in some CIFAR-10 images, and this small change sometimes caused classification mistakes with high output scores, employing well-known models such as VGG [31] and AlexNet [32]. This result increased the safety concerns in critical applications.

Approaches using Bayesian modeling have shown good performance during uncertainty analysis of deep learning models. For instance, the Bayesian Neural Network (BNN) models [33,34,35,36,37] introduce stochastic components over the network parameters θ, simulating various possible models with their probability distribution $p\left(\theta \right)$. Although these architectures are computationally more expensive than deterministic ones, they offer prediction uncertainty and consequently more reliable outputs.

This work introduces two BNN architectures termed SCBNN and SCBNN-MOPED for MI classification and uncertainty quantification, which were trained here by using the variational inference procedure. In our study, the SCBNN-MOPED architecture uses the MOPED method from [38] to establish the prior distribution of each weight by employing the weights of the pre-trained equivalent deterministic network. The SCBNN architecture initializes the prior distribution of each weight as a standard Gaussian distribution $N\left(0,1\right)$, which is established by default in the Python’s TensorFlow Probability package.

As another novelty, an adaptive threshold scheme is also introduced here to implement the MI classification with a reject option. As an advantage, this scheme uses h a closed formula to calculate the threshold, rather than performing an exhaustive search over the space of possible threshold values as performed in previous works [39,40,41]. The datasets 2a and 2b from the BCI Competition IV [42] were used for evaluation, considering both subject-specific and non-subject-specific strategies for training. As a result, our approach improved the MI classification with/without a reject option. In terms of efficacy, this proposed threshold scheme performed better with respect to other approaches that estimate the best threshold value by using an uncertainty metric.

The main contributions of this current research are listed as follows:Two BNN architectures by applying the variational inference method for MI classification are proposed, confirming the advantage of using MOPED’s prior distribution for Bayesian models. Additionally, an efficient implementation to reduce computational cost for practical real-world applications is proposed.An adaptive threshold scheme based on a closed formula for robust MI classification with a reject option is proposed here.

## 2. Materials and Methods

### 2.1. Datasets Description

The two popular datasets 2a and 2b from BCI Competition IV were used.

The dataset 2a contains EEG signals with a sampling rate at 250 Hz, recorded during four MI tasks (left hand, right hand, tongue, and foot) from nine healthy subjects over 22 locations (Fz, FC1, FC3, FCz, FC2, FC4, C5, C3, C1, Cz, C2, C4, C6, CP3, CP1, CPz, CP2, CP4, P1, Pz, P2 and POz). Two sessions on different days were collected for each subject, completing a total of 288 trials per session. In our study, the first and second sessions were used for training and testing, respectively, as in [13,43,44,45,46,47].

The dataset 2b comprises EEG signals collected during the execution of two MI tasks (left hand and right hand) from nine subjects with three bipolar EEG channels (around C3, Cz, and C4), using a sampling rate at 250 Hz. Each subject completed five sessions with a total of 720 trials, comprising 120 trials per session in the first two sessions and 160 trials per session in the remaining three sessions. In our research, the first three sessions were used for training, and the last two sessions for testing, as in [15,16,17,18,44,48].

### 2.2. Preprocessing and Data Augmentation

Previous studies [49,50,51,52,53,54] reported that real or imagined unilateral movement enhances the mu (8 to 12 Hz) and beta (13 to 30 Hz) rhythms over the primary motor cortex in both contralateral and ipsilateral hemispheres, respectively; phenomena known as event-related desynchronization/synchronization (ERD/ERS) [55]. Thus, the EEG signals of each trial were band-pass filtered in our study with a frequency range from 4 to 38 Hz through a fourth-order Butterworth filter [13,14,45,47,56], aiming to preserve the ERD and ERS potentials, rejecting noise and undesirable physiological and non-physiological artifacts. Afterwards, each filtered EEG trial **x** was standardized as $\left(x\left(t_{i}\right)-\mu \left(t_{i}\right)\right)/\sigma \left(t_{i}\right)$ by applying the electrode-wise exponential moving standardization with a decay factor of 0.999, which computes both mean $\mu \left(t_{i}\right)$ and variance $\sigma^{2}\left(t_{i}\right)\;$ values taken at sample $t_{i}$ [45]. The starting values $\mu \left(t_{0}\right)$ and $\sigma^{2}\left(t_{0}\right)$ were calculated over periods corresponding to the rest state preceding each trial. In order to rectify outliers, the EEG amplitudes of each trial were first limited to $\;\mu \left(t_{i}\right)\pm 6\sigma \left(t_{i}\right)$. Finally, the trial crop strategy for data augmentation was employed. For both datasets, crops of 4 s in length each 8 ms from −0.5 to 4 s (cue onset at 0 s) over all trials were extracted in our study. Figure 1 shows the EEG preprocessing and data augmentation.

### 2.3. Baseline Architecture

A shallow architecture based on a Convolutional Neural Network (SCNN) [57] was used in our work as a baseline deterministic network, as shown in Figure 2. This architecture contains two convolutional layers and a dense classification layer. The first convolution has an input tensor of 1000×c×1, where *c* is the number of EEG channels. The first convolution applies a temporal convolution with a 45×1 filter and 40 channels, giving an output tensor of 478×c×40 after performing a down-sampling with a stride of 2. The second convolutional layer realizes a spatial convolution composed of 40 channels and a 1×c filter, which performs a convolution amongst all EEG channels. Following the spatial convolution, the model executes a sequence of transformations as follows: a batch normalization layer, an activation function with square non-linearity, an average pooling layer with 45×1 sliding windows, and a max-pooling layer with 8×1 pool size, 8×1 stride, and a logarithmic activation function. Lastly, the classification layer composed of a dense layer using the Softmax activation function receives 2160 features, which are translated into a s×1 prediction vector, s being the number of classes.

Although dropout layers and the “MaxNorm” regularization have been previously used in the baseline architecture [57] to avoid overfitting, we did not use them in BNN models that are already more robust and less prone to overfitting. Furthermore, the “Early Stopping” and the decay of learning rate techniques were used. Additionally, the Adam optimizer and the Categorical Cross-Entropy as a cost function were employed.

### 2.4. Bayesian Neural Networks

BNNs [58] provide a probabilistic interpretation of deep learning models by placing distributions over their weights. In BNNs, the weights are no longer considered as fixed values, but rather as random variables sampled according to a distribution whose parameters are learned in the training stage. This makes each prediction different for the same input, and for many forward passes, the average behavior produces relevant results. Notably, the variability of these predictions also allows assessment of the model’s confidence.

Let D=X, Y be a training set with inputs X=x1, ⋯, xn and expected outputs or targets Y=y1, ⋯, yn, where  xn∈ℝd and  yi∈ℝC, C being the number of classes. The Bayesian modeling aims to approximate the posterior distribution of weights pw|D that generated the output vector Y. Following this approach, the prior distribution pw that represents the initial beliefs about the weights has to be established.

If $p\left(w|D\right)$ is known, then for a given input $\hat{x}$, the predictive distribution $p(\hat{y}\;|\;\hat{x},\;D)$ can be determined from the outputs $p(\hat{y}\;|\;\hat{x},\;w)$ corresponding to a specific set of weights w as follows:(1)$$p(\hat{y}\;|\;\hat{x},\;D)=\int p(\hat{y}|\;\hat{x},\;w)\;p(w|D)dw$$

The integral in (1) can be estimated by using the Monte Carlo method. For this, it is necessary to perform T evaluations of the neural network on the same input $\hat{x}$ and weights $w_{t}$ sampled from the posterior distribution  p(w|D). As a result, instead of a single output, we obtain T outputs from the model $\;\left\{{\hat{y}}_{t};1\;\leq t\leq T\right\}$. According to [21], T=50 is a good balance in terms of complexity and accuracy. The set $\left\{{\hat{y}}_{t}\right\}$ can be interpreted as a sample of the predictive distribution. The final prediction of BNN on the input $\hat{x}$ is seen as the sample mean of the predictive distribution:(2)$$E\left[p\left(\hat{y}\right)\right]\approx \frac{1}{T}\overset{T}{\underset{t=1}{\sum }}p\left(\hat{x},\;w_{t}\right)=\frac{1}{T}\overset{T}{\underset{t=1}{\sum }}{\hat{y}}_{t};\;\;\;\;\;\;\;\;\;w_{t}\sim \;p(w|D)$$

In neural networks, the uncertainty measures how reliable a model is when making predictions. The statistical dispersion of the predictive distribution $p(\hat{y}\;|\;\hat{x},\;D)$ is one indicator of uncertainty. A “large” dispersion in $p(\hat{y}\;|\;\hat{x},\;D)$ means that the neural network produces less confident results. In contrast, a “small” dispersion indicates that the network always provides similar results independently of the sampled weights $w_{t\;}$.

To summarize, if the posterior distribution is known, it is possible to obtain, for each input $\hat{x},$ an approximation of the predictive distribution $\;p(\hat{y}\;|\;\hat{x},\;D)$. This then allows us to estimate the BNN prediction and an uncertainty measure. However, finding and sampling the posterior distribution for complex models, such as neural networks, is a computationally intractable problem because of their high dimensionality and non-convexity. To address this issue, two popular approaches have been introduced previously in other studies, such as (1) variational inference [33] and (2) Monte Carlo dropout [21]. The next section only describes the former method, which was used in our approach.

### 2.5. Variational Inference

In variational inference, rather than sampling from the exact posterior distribution p(w|D), a variational distribution q(w, θ) is used, parametrized by a vector θ. The θ values are then learned in such a way that q(w, θ) is close as possible to p(w|D) in a Kullback–Leibler (KL) divergence sense. It is known that minimizing the KL divergence between q(w, θ) and $p\left(w|D\right)$ is equivalent to maximizing the evidence lower bound (ELBO) function, denoted as $L\left(\theta \right)$, which serves as the objective function to train the model.
(3)$$L\left(\theta \right)=\int q\left(w,\;\theta \right)\;\ln\;p(Y|X,\;w)dw-D_{KL}\left(q\left(w,\;\theta \right),p\left(w\right)\right)$$

Maximizing the first term in (3) encourages $q\left(w,\;\theta \right)$ to explain the data well, while minimizing the second term encourages $q\left(w,\;\theta \right)$ to be close to $p\left(w\right)$.

Although in general guessing $q\left(w,\;\theta \right)$ is complex, to adopt an independent Gaussian distribution $w∼N\left(\mu ,\;\sigma^{2}\right)$ for each weight is a simple and common approach that often works. Additionally, with respect to the prior distribution, a standard normal distribution $w∼N\left(0,\;1\right)$ is commonly used. In contrast, the MOPED method [38] proposes to specify prior distributions $p\left(w\right)$ from deterministic weights $\left(w_{d}\right)$ derived from a pretrained DNN with the same architecture. 

### 2.6. Bayesian Neural Models for Uncertainty Estimation and MI Classification

In this work, two BNN architectures (SCBNN and SCBNN-MOPED) derived from the SCNN baseline are proposed. They only differ in the way the prior distribution of the weights is established. The former initializes the prior distribution of weights as a standard Gaussian distribution $N\left(0,1\right)$, whereas the second uses the MOPED method. For SCBNN-MOPED, the prior distribution of each weight w is initialized as an independent Gaussian distribution with the mean μ taken from the pretrained weights $w_{d}$ of the SCNN architecture and a scale $\sigma =0.1\left|w_{d}\right|.$

Both models were obtained by starting from a deterministic network SCNN and translating it onto a Bayesian network by using the Flipout estimator [59] from TensorFlow. We preferred the Flipout estimator instead of using reparameterization [60] because it offers a significant low variance, albeit it uses roughly twice that of floating point operations. Table 1 shows the main differences between SCNN and the two proposed Bayesian neural architectures, following the terminology used in the “TensorFlow probability” library of Python for better comparison [33].

### 2.7. Uncertainty Measures

Several uncertainty measures can be used to quantify the model uncertainty [33,39,40,61]. For better understanding, three well-known metrics—predictive entropy $\left(ℍ\right)$, mutual information $\left(I\right)$, and margin of confidence $\left(M\right)$—were first presented as follows.

Let C be the total number of classes and $\;{\hat{y}}_{t}=\left({\hat{y}}_{1t},\;{\hat{y}}_{2t},\;\cdots ,\;{\hat{y}}_{Ct}\right)$ the model output for a given input $\hat{x}$ in a stochastic forward pass t; 1≤t≤T. Let ${\hat{y}}^{*}=\left({\hat{y}}_{1}^{*},{\hat{y}}_{2}^{*},\cdots ,\;{\hat{y}}_{C}^{*}\right)$ be the average of predictions $\left\{{\hat{y}}_{t};1\leq t\leq T\right\}$.

**Predictive Entropy** ℍ. This metric captures through Equation (4) the average on the amount of information contained in the predictive distribution, reaching its maximum value $\left({log}_{2}C\right)$ when all classes have equal uniform probability (maximum uncertainty, in other words). In contrast, it reaches zero value (the minimum) when a unique class has probability equal to 1, confirming a certain prediction.
(4)$$ℍ\approx -\overset{C}{\underset{j=1}{\sum }}{\hat{y}}_{j}^{*}\log_{2}({\hat{y}}_{j}^{*})$$

To facilitate the comparison across various datasets, the predictive entropy was normalized as follows: ${ℍ}_{n}=ℍ/{log}_{2}C$, ${ℍ}_{n}\in \left[0,1\right]$.

**Mutual Information** I. This measures the epistemic uncertainty by capturing the model’s confidence from its output.
(5)$$I\approx \;ℍ-\frac{1}{T}\overset{T}{\underset{t=1}{\sum }}\overset{C}{\underset{j=1}{\sum }}{\hat{y}}_{jt}{log}_{2}({\hat{y}}_{jt})$$

**Margin of Confidence** M. Let $c=argmax\;{\hat{y}}_{j}^{*}$ be the predicted class. The most intuitive form of measuring uncertainty is through analyzing the difference between two predictions of highest confidence. For this, in each forward pass t, the difference $d_{t}$ between the output ${\hat{y}}_{ct}$ (predicted class) and the other output of highest score value $\underset{j\neq c}{max}{\hat{y}}_{jt}\;$ is calculated. These differences are then averaged as follows:(6)$$M=\frac{1}{T}\overset{T}{\underset{t=1}{\sum }}d_{t};\;\;\;\;\;\;\;\;\;\;d_{t}={\hat{y}}_{ct}-\underset{j\neq c}{max}{\hat{y}}_{jt}$$

### 2.8. Classification with a Reject Option

The uncertainty values, calculated by using any of the aforementioned measures, provide to classifiers the ability to accept or reject inputs. For instance, a high uncertainty means that the classifier may be performing random predictions; therefore, the associated input should be rejected. This type of classification is known as classification with a reject option [36,62], which is of great importance for applications that require low error tolerance. Classification with a reject option generalizes the decision problem of class predictions, and also determines whether the prediction is reliable (accept) or unreliable (reject).

The process of refraining from providing a response or discarding an input when the system does not have enough confidence in its prediction is a topic of interest that has been addressed over the last 60 years [63]. Recent methods [39,40,41] have implemented the classification of rejection from an established threshold Th by using any uncertainty metric, rejecting inputs that present an uncertainty value above Th.

The criteria used in [39] incorporate the ground-truth label, model prediction, and uncertainty value to evaluate the selected threshold. In this approach, the map of *correct* and *incorrect* values is computed by matching the ground truth labels with the model’s predictions. Then, given a threshold Th, the predictions are classified as *certain* and *uncertain*, providing four combinations to predict an input as follows: incorrect–uncertain (iu), correct–uncertain (cu), correct–certain (cc), and incorrect–certain (ic).

Let $\;N_{iu}$,$\;N_{cu},\;N_{cc}$ and $N_{ic}$ be the total number of each type of predictions in each subset, where N is the total number of predictions, and $R_{c}$ is the proportion of certain predictions with respect to the total number of predictions:(7)$$R_{c}\left(Th\right)=\frac{P\left(\mathrm{certain}\right)}{N}=\frac{N_{cc}+N_{ic}}{N}$$

Then, the following criteria measure the effectiveness of an uncertainty estimator and a threshold selector [39]:

The correct–certain ratio $R_{cc}$ measures the ability of a classifier to accurately classify non-rejected samples:(8)$$R_{cc}\left(Th\right)=\frac{P\left(\mathrm{correct}\cap \mathrm{certain}\right)}{P\left(\mathrm{certain}\right)}=\frac{N_{cc}}{N_{cc}+N_{ic}}$$

The overall accuracy of the uncertainty estimation can be measured through the Uncertainty Accuracy $\left(UA\right)$, which indicates the ability of a classifier to accurately classify non-rejected samples, and reject misclassified samples:(9)$$UA\left(Th\right)=\frac{N_{cc}+N_{iu}}{N}=1-\frac{N_{ic}+N_{cu}}{N}$$

$UA\left(Th\right)$ penalizes both the incorrect–certain and correct–uncertain predictions, and its highest values suggest better thresholds. Thus, this criterion can be also used to compare between different threshold values, in order to increase the reliability, effectivity, and feasibility for practical applications.

In this study, the correct–uncertain ratio $R_{cu}$ was proposed to calculate the accuracy of rejected predictions, as shown in Equation (10). This criterion was incorporated with the objective of evaluating how close the accuracy on rejected predictions was with respect to the hypothetical accuracy that would be obtained if the classifier provided random predictions.
(10)$$R_{cu}\left(Th\right)=\frac{P\left(\mathrm{correct}\cap \mathrm{uncertain}\right)}{P\left(\mathrm{uncertain}\right)}=\frac{N_{cu}}{N_{cu}+N_{iu}}$$

### 2.9. Adaptive Uncertainty Threshold

To select the cut-off threshold, recent works [39,40,41] propose to evaluate all threshold values for a given uncertainty measure, such as predictive entropy. For this purpose, this study measured the performance by using each threshold, taking into account a criterion of quality as UA, consequently keeping the best threshold. This strategy has high computational cost because of its expensive and exhaustive search. As a contribution, we introduced a low-computational-cost novel threshold scheme based on a closed formula, rather than an exhaustive search.

Our threshold scheme uses Equation (6) to compute the margin of confidence M, which takes values close to zero when the prediction has high uncertainty. It is worth mentioning that if M is not significantly higher than zero, there will be no significant difference between the two highest confidence predictions of the model. Therefore, we reduced the labeling problem on the prediction of an input $\hat{x}$ as uncertain to a hypothesis test for the mean $E\left[D\left(\hat{x}\right)\right]$, where $D\left(\hat{x}\right)$ denotes all possible differences $\left\{d_{t}={\hat{y}}_{ct}-\underset{j\neq c}{max}{\hat{y}}_{jt}\right\}$ that could be obtained from the forward passes associated with the input $\hat{x}$. As a result, if the prediction is certain, the null hypothesis $H_{0}$: $E\left[D\left(\hat{x}\right)\right]\leq 0$ must be rejected, whereas the alternative hypothesis $H_{1}$: $E\left[D\left(\hat{x}\right)\right]>0$ must be accepted. The test statistic is:(11)$$\zeta =\frac{M\;}{\sigma_{d}/\sqrt{T}}$$
where $\sigma_{d}$ is the standard deviation of the sample ${\left\{d_{t}\right\}}_{t=1}^{T}$, and T is the number of forward passes over each input $\hat{x}$. It is worth remarking that the rejection region for the null hypothesis is the right tail of the distribution.

M is a sample mean of T=50 random values ${\left\{d_{t}\right\}}_{t=1}^{T}$, making M follow a normal distribution with variance $\sigma_{d}^{2}/T$. Therefore, we assume that the random variable ζ follows a standard normal distribution. The quantile $z_{1-\alpha }=\Phi^{-1}\left(1-\alpha \right)$ is used as the critical value of the test, where Φ is the cumulative distribution function (CDF) of the standard normal distribution, and 1−α is the confidence level. The null hypothesis is rejected for $\zeta >z_{1-\alpha }$. Thus, if the prediction is certain, M must be greater than $\sigma_{d}z_{1-\alpha }/\sqrt{T}$. This way, the proposed adaptive threshold can be calculated as follows:(12)$$T_{M}\left(\hat{x}\right)=\frac{\sigma_{d}z_{1-\alpha }}{\sqrt{T}}$$

As a highlight, our adaptive threshold scheme highlights four observations. First, the threshold scheme is statistically based on a closed formula. Second, the threshold is not fixed; it varies depending on the sample variance $\sigma_{d}^{2}$ of ${\left\{d_{t}\right\}}_{t=1}^{T}$. In other words, this novel threshold is adaptive to the predictive distribution characteristics of each input $\hat{x}$. Third, our scheme is conservative, as it does not classify as uncertain those inputs in which the predictions for each forward pass are consistent ($\sigma_{d}\approx 0$). However, increasing the confidence level 1−α, the threshold scheme behaves less conservatively. Fourth, our threshold scheme can be considered as universal, as it does not require prior knowledge about the data, depending exclusively on the predictive distribution.

### 2.10. Experiments

The proposed methods were evaluated with three experiments, using subject-specific and non-subject-specific classification strategies.

The first experiment analyzed the accuracy of the Bayesian Neural Network models SCBNN and SCBNN-MOPED for MI classification, compared with their deterministic counterpart SCNN [57].

The second experiment assessed the quality using the proposed adaptive threshold scheme within the SCBNN-MOPED model. The quality was measured by looking into accepted and rejected partitions of the testing set, measuring how well each partition could be classified. Ideally, the rejected partition should be hard to classify, with significantly better accuracy than random guessing.

The last experiment evaluated the proposed threshold scheme with respect to [39] as a benchmark, as it seeks the optimum threshold via exhaustive search. The aim was to demonstrate that our proposed adaptive threshold may achieve the state-of-the-art performance without brute force.

All experiments were designed to meet the requirements for the BCI Competition IV dataset, never using the testing sets during training or validation. Figure 3 shows the difference between both subject-specific and non-subject-specific strategies, respectively, where the former restricts itself to the training data of each subject (see Figure 3a), whereas the latter calibrates the classifier by using the training data from all subjects (see Figure 3b).

In all experiments, a repeated holdout validation over the same testing set was carried out, considering both subject-specific and non-subject-specific classification strategies. For each repetition, a subset $V_{i}$ of the training set T was randomly picked, guaranteeing identical class ratios in $V_{i}$ with respect to the class ratios in T. The subset $V_{i}$ was used for validation purposes, which is necessary to build deep learning models. For training purposes, the remaining set $T_{i}=T\;\backslash \;V_{i}$ was used. Once the model was trained for each pair $T_{i}$ and $V_{i}$, it was evaluated on the testing set E. To reduce the influence of random factors, this process was repeated 16 times for each model, reporting the average of some metrics, such as accuracy, $\;R_{cc}$, among others.

Specifically, the accuracy was used for evaluation in the first experiment, whereas the criteria $R_{c}$, $R_{cc}$ and $R_{cu}$ (see Equations (7), (8) and (10)) were used in the second experiment, assessing the quality of accepted and rejected partitions on the testing set. As in [39], we used UA (Equation (9)) as an evaluation criterion in the third experiment. 

For the third experiment, the method in [39] was replicated by using the SCBNN-MOPED architecture and the normalized predictive entropy ${ℍ}_{n}$ as the uncertainty measure. As in [39,40,41], the optimum threshold was obtained by making the uncertainty threshold Th vary over the interval [0.05, 1] and calculating UA over the validation set $V_{i}\;$ for each Th. Once the optimal threshold was found, its performance was evaluated with respect to the criterion UA by using the testing set E. Similar to the other experiments, 16 validation sets ($V_{1}\;\cdots ,\;V_{16}$) were used to compute the overall assessment of the method in [39].

An HPC using a Dell PowerEdge R720 server with four 2.40 GHz Intel Xeon processors and 48 GB was used for all experiments. An NVIDIA GM107L Tesla M10 GPU with 32 GB memory was used to train and test the Bayesian models by using the Python TensorFlow 2.3.0 framework.

## 3. Results and Discussion

### 3.1. MI Classification Using SCBNN and SCBNN-MOPED Models

Table 2, Table 3, Table 4 and Table 5 show the results for subject-specific and non-subject specific strategies by using the datasets 2a and 2b, respectively, where a confidence level 1−α at 0.95 by using the Wilcoxon signed rank test was considered.

For both datasets, the SCBNN-MOPED architecture achieved significantly better results than the SCBNN for almost all subjects. For dataset 2a, the accuracy improvement was greater than 22% and 19% (on average) in the subject-specific and non-subject-specific strategy, respectively. In general, our results were consistent with the findings in [38], showing that the MOPED method helped the training convergence and improved the accuracy of Bayesian neural models.

The SCBNN-MOPED model also achieved better mean accuracy than the deterministic SCNN model, except for the subject-specific classification in dataset 2a. Table 4 and Table 5 show that, for non-subject-specific classification, SCBNN-MOPED significantly outperformed the SCNN model with respect to overall accuracy, and for almost all subjects.

Because the SCBNN-MOPED architecture achieved superior performance, the next experiments were carried out discarding the SCBNN architecture.

### 3.2. Classification with Reject Option Using SCBNN-MOPED Architecture

Figure 4 and Figure 5 show the performance by applying the proposed adaptive threshold for MI classification with a reject option, considering both subject-specific and non-subject-specific classifications, respectively.

Figure 4 and Figure 5 show that a larger number of predictions were rejected in dataset 2a independently of the training strategies. As expected from an adaptive and effective threshold, the accuracy achieved on the accepted predictions was higher than considering the full testing set without rejection. This was notable, using $R_{cc}$ as criterion for comparison with the results shown in Table 2, Table 3, Table 4 and Table 5. This supports that by rejecting uncertain predictions, the classification with a reject option can improve both accuracy and reliability, which is of great importance in systems for real-life applications.

The $R_{cu}$ criterion is another way to measure the quality of our proposed threshold scheme, looking into the accuracy obtained over the rejected predictions. For dataset 2b and both training strategies, $R_{cu}$ of 50% was achieved, showing that the classification accuracy over rejected predictions and random guessing was not different, confirming the optimal performance using the proposed threshold scheme. For dataset 2a, $R_{cu}$ of 40% was obtained, while a random classifier would obtain 25%.

It is worth noting that our approach based on an adaptive threshold is conservative, achieving significant low percentage of rejected predictions in comparison to the incorrect predictions. For instance, 33.23% of inputs were incorrectly classified in dataset 2a with a non-subject-specific strategy, and only 1.22% was associated as uncertain by our adaptive threshold. However, by increasing the confidence level 1−α, the number of rejected predictions can be increased. This strategy has limitations, though, as our proposed threshold scheme only considers epistemic uncertainty, and it may also overlook other sources of uncertainty.

### 3.3. Comparison with Other Methods Based on Thresholds

Table 6 and Table 7 show the UA values obtained by our proposed adaptive threshold $\left(UA\left(T_{M}\right)\right)$, as well as the maximum or optimal values reached by [39] via exhaustive search and normalized predictive entropy $\left(U,A,\left(T_{ℍ}\right)\right)$.

Recall that the UA criterion makes it possible to evaluate different cut-off thresholds. By inspecting the values in Table 6 and Table 7, we conclude that our proposed threshold scheme allows us to achieve state-of-the-art performance. Moreover, by relying exclusively on the predictive statistical distribution, our scheme becomes universal and independent of the nature of data.

## 4. Considerations for Reducing Computational Cost

All evaluations considering the classification accuracy and uncertainty analysis throughout this current work were carried out by using data augmentation, as shown in Figure 1. The accuracy was obtained as a ratio between correct classified crops and the total crops, as proposed in the CAgross method [19]. Others studies followed different strategies. For instance, a single prediction for each trial was obtained in the CAST method [19] by using majority voting, whereas in [45] the authors averaged the predictions over all crops. This analysis of the crops requires an extra computational cost, which increases even more with the use of Bayesian architectures, as it realizes *T* forward passes to obtain the prediction over the same input.

Given the nature of the problem that is addressed in our study, the EEG variability of patterns associated with MI tasks, and the large intra- and inter-subject differences can be observed through the ERD/ERS potentials throughout. As a result, in certain crops, the aforementioned phenomena were not detected appropriately by the model, producing incorrect predictions. Figure 6a shows the predictive distribution histograms of crops over a specific trial on Subject A01 from dataset 2a, where the initial crops were classified correctly as left-hand MI, but after approximately crop number 45, they were incorrectly predicted as right-hand MI. Figure 6b shows that this change correlated with the increase of uncertainty. Observing this behavior, we believe that, when considering the predictions over all crops predicting the MI of each trial, it is not the most appropriate strategy, since all crops do not have the same certainty. Therefore, it is preferable to consider the crops with less uncertainty.

We hypothesize that if the predictions with less uncertainty are determined over the same time interval for all trials on the validation set, similar behavior may be also obtained in the testing set. This allows us to choose only crops contained in that interval for classification with/without a reject option, reducing the computational cost with high accuracy.

Figure 7 shows the normalized predictive entropy on a validation set from database 2a. We observed that the central crops covering a time interval from 1.724 to 5.756 s regularly presented the lowest uncertainty with respect to the crops located at the extremes. It is worth mentioning that this finding was also observed in dataset 2b.

From our findings, we selected the five central crops of each trial, aiming to reduce computational cost during the evaluation step. Then, we averaged the predictions of these five central crops to rank each trial. Table 8 shows the classification accuracy without/with a reject option achieved on the selected crops of each trial (from the testing set), using SCBNN-MOPED architecture and subject-specific strategy with each database. For a better assessment of the established criterion, we also added the achieved results from Table 2 and Table 3, using all the crops.

The results show that the established criterion to reduce the computational cost by using the five central crops provides advantages during MI classification without/with a reject option. This proposal increases the possibilities of real implementations using Bayesian classifiers for MI tasks. However, for practical implementation, other criteria must be considered, since neural network models require large hardware resources. This analysis is outside the scope of this current work.

## 5. Conclusions

The advantage of using the MOPED method to improve MI classification by employing BNN has been verified in this work. The uncertainty quantification on predictions by using the margin of confidence provided valuable information, which enhanced the MI classification. In addition, the proposed adaptive threshold scheme allowed us to obtain a more robust, effective and adequate classifier. The proposed scheme, formulated in a closed equation, is based exclusively on the predictive statistical distribution, which makes it a universal method. Thus, it can be extended to other classification problems. Finally, a criterion to reduce the computational cost based on the uncertainty analysis was proposed, which increases the possibilities of practical implementations of Bayesian classifiers in MI-based BCIs.

## Figures and Tables

**Figure 1 sensors-23-00703-f001:**
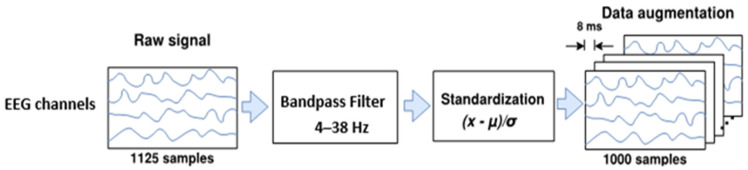
The raw EEG preprocessing and data augmentation.

**Figure 2 sensors-23-00703-f002:**
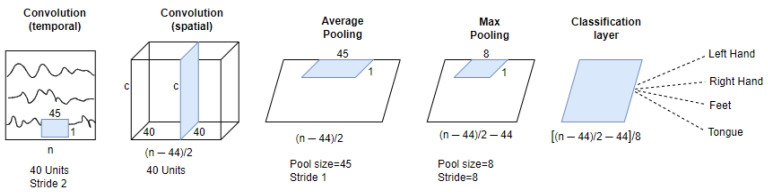
Shallow Convolutional Neural Network as baseline architecture.

**Figure 3 sensors-23-00703-f003:**
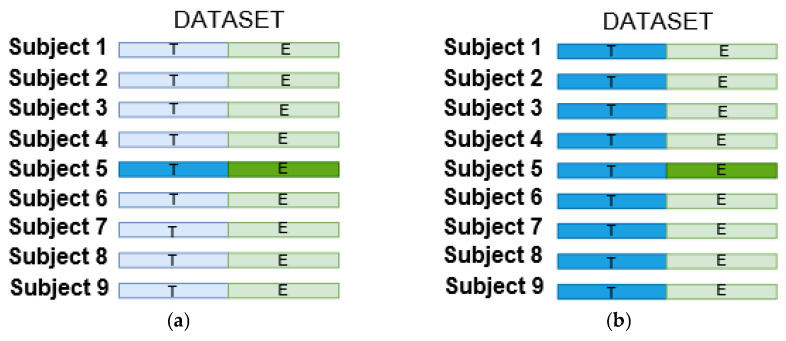
The training ( T in blue) and testing ( E in green) set selections for both strategies: (**a**) subject-specific; (**b**) non-subject-specific.

**Figure 4 sensors-23-00703-f004:**
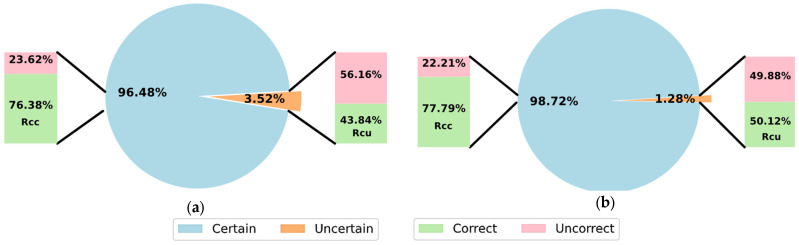
Results using SCNN-MOPED with the adaptive threshold scheme for MI classification with reject option, considering the subject-specific strategy, (**a**) dataset 2a; (**b**) dataset 2b.

**Figure 5 sensors-23-00703-f005:**
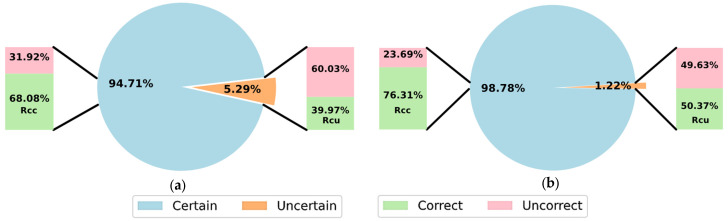
Results using SCNN-MOPED with the adaptive threshold scheme for MI classification with reject option, considering the non-subject-specific strategy, (**a**) dataset 2a; (**b**) dataset 2b.

**Figure 6 sensors-23-00703-f006:**
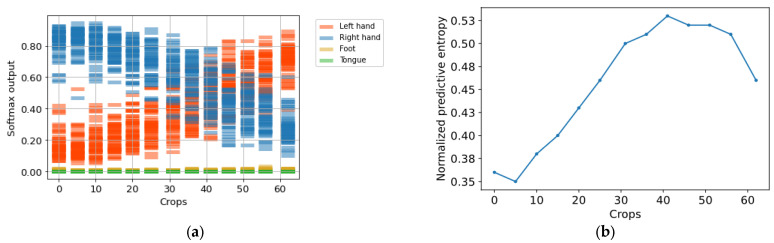
(**a**) Predictive distribution histograms of crops; (**b**) normalized predictive entropy of crops.

**Figure 7 sensors-23-00703-f007:**
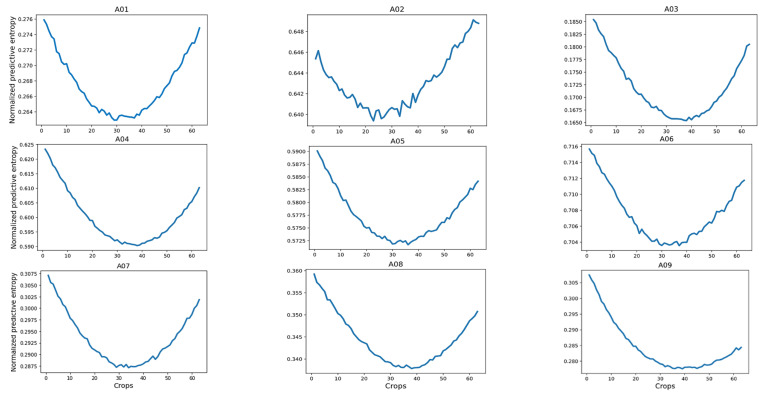
Average of the normalized predictive entropy for each crop on subjects in dataset 2a.

**Table 1 sensors-23-00703-t001:** Comparison between SCNN and the two proposed Bayesian neural architectures.

Layer	SCNN	SCBNN and SCBNN-MOPED
Temporal Convolution	Conv2D	Conv2DFlipout
Spatial Convolution	Conv2D	Conv2DFlipout
Dropout Layer	Yes	No
Classification Layer	Dense	DenseFlipout

**Table 2 sensors-23-00703-t002:** Subject-specific classification results (% accuracy) on dataset 2a.

Model	A01	A02	A03	A04	A05	A06	A07	A08	A09	Mean
**SCNN** [57]	**83.81**	**51.97**	91.48	73.82	69.82	53.90	**91.17 ***	**81.87 ***	**82.39 ***	**75.58**
**SCBNN**	62.95	27.48	78.90	41.34	35.75	36.95	56.12	66.27	68.92	52.74
**SCBNN-MOPED**	83.64 **^†^**	51.72 **^†^**	**91.50 ^†^**	**74.53 ^†^**	**70.84 ^†^**	**55.28 ^†^**	89.42 **^†^**	80.63 **^†^**	81.12 **^†^**	75.41 **^†^**

(*) indicates significant differences with 95% confidence level; (^†^) indicates significant differences between the SBBNN and SCBNN-MOPED.

**Table 3 sensors-23-00703-t003:** Subject-specific classification results (% accuracy) on dataset 2b.

Model	B01	B02	B03	B804	B05	B06	B07	B08	B09	Mean
**SCNN** [57]	75.43	**55.36**	52.09	94.96	**87.60**	79.71	**79.77**	87.87	**84.69 ***	77.50
**SCBNN**	69.92	50.46	51.53	94.56	84.23	78.87	77.76	86.10	79.13	74.73
**SCBNN-MOPED**	**75.97 ^†^**	54.41 **^†^**	**52.36**	**95.44 ***	87.38 **^†^**	**80.38 ^†^**	79.37 **^†^**	**88.52 ^†^**	83.95 **^†^**	**77.53 ^†^**

(*) indicates significant differences with 95% confidence level; (^†^) indicates significant differences between the SBBNN and SCBNN-MOPED.

**Table 4 sensors-23-00703-t004:** Non-subject-specific classification results (% accuracy) on dataset 2a.

Model	A01	A02	A03	A04	A05	A06	A07	A08	A09	Mean
**SCNN** [57]	72.29	39.26	81.59	60.90	**54.03 ***	51.58	**74.70 ***	**77.11**	**76.86**	65.37
**SCBNN**	60.14	29.19	63.85	40.87	32.98	38.57	40.61	62.42	58.89	47.50
**SCBNN-MOPED**	**82.44 ***	**47.42 ***	**83.25 ***	**62.31 ***	49.73 **^†^**	**52.79 ^†^**	70.53 **^†^**	76.67 **^†^**	75.74 **^†^**	**66.77 ***

(*) indicates significant differences with 95% confidence level; (^†^) indicates significant differences between the SBBNN and SCBNN-MOPED.

**Table 5 sensors-23-00703-t005:** Non-subject-specific classification results (% accuracy) on dataset 2b.

Model	B01	B02	B03	B04	B05	B06	B07	B08	B09	Mean
**SCNN** [57]	66.94	55.66	**54.08 ***	93.78	**79.09**	80.81	**74.18 ***	90.05	**83.42**	75.33
**SCBNN**	69.53	56.07	52.69	92.81	74.98	78.60	72.75	89.48	80.58	74.17
**SCBNN-MOPED**	**73.32 ***	**57.22 ***	53.10	**94.08 ***	78.57 **^†^**	**81.52 ***	73.20	**90.38**	83.15 **^†^**	**76.06 ***

(*) indicates significant differences with 95% confidence level; (^†^) indicates significant differences between the SBBNN and SCBNN-MOPED.

**Table 6 sensors-23-00703-t006:** Comparison between the proposed adaptive threshold and the method from [39], using subject-specific classification.

Subject	Dataset 2a		Dataset 2b	
	$UA(T_{M})$	$UA\left(T_{H}\right)$	$UA\left(T_{M}\right)$	$UA\left(T_{H}\right)$
01	**83.58**	83.27	**75.95**	75.70
02	53.62	**55.41 ***	**54.37 ***	53.17
03	**91.50**	90.95	**52.34 ***	51.04
04	**75.41**	75.08	**95.44**	95.11
05	**71.99**	71.63	**87.39**	86.66
06	56.75	**60.21 ***	**80.39**	80.16
07	**89.41**	88.03	**79.38**	79.16
08	**80.92**	80.66	**88.52**	87.89
09	81.28	**81.94**	**83.91 ***	82.23
**Mean**	76.05	**76.35**	**77.52 ***	76.79

(*) indicates significant differences with 95% confidence level.

**Table 7 sensors-23-00703-t007:** Comparison between the proposed adaptive threshold and the method from [39], using non-subject-specific classification.

Subject	Dataset 2a		Dataset 2b	
	$UA(T_{M})$	$UA\left(T_{H}\right)$	$UA\left(T_{M}\right)$	$UA\left(T_{H}\right)$
01	**82.46**	82.07	73.30	**73.32**
02	51.50	**54.01 ***	57.18	**57.23**
03	**83.62**	83.54	53.08	**53.10**
04	63.85	**65.46 ***	94.08	**94.09**
05	51.97	**55.89 ***	78.58	78.58
06	54.23	**56.39 ***	81.49	**81.54**
07	**71.11 ***	69.55	**73.21**	73.20
08	77.36	**77.68**	90.38	90.38
09	76.25	**76.60**	83.14	**83.16**
**Mean**	68.04	**69.02 ***	76.05	76.07

(*) indicates significant differences with 95% confidence level.

**Table 8 sensors-23-00703-t008:** Performance of SCBNN-MOPED (% accuracy) by using the selected crops for subject-specific classification.

	2a			2b		
Subject	All Crops	Five Central Crops		All Crops	Five Central Crops	
		Without Rejection	With Rejection		Without Rejection	With Rejection
01	83.64	84.27	84.42	75.97	76.07	76.23
02	51.72	52.50	53.08	54.41	54.96	54.91
03	91.50	91.95	92.07	52.36	52.32	52.28
04	74.53	75.91	76.61	95.44	95.59	95.60
05	70.84	71.31	72.28	87.38	87.15	87.31
06	55.28	56.36	56.87	80.38	80.96	81.14
07	89.42	90.04	90.32	79.37	79.69	79.80
08	80.63	81.49	81.73	88.52	88.73	88.90
09	81.12	82.25	82.51	83.95	84.32	84.55
**Mean**	75.41	76.23	76.65	77.53	77.75	77.85

## Data Availability

Not applicable.

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
