# Peer review of "Robust Motor Imagery Tasks Classification Approach Using Bayesian Neural Network"

_sensors, 2023, doi:10.3390/s23020703_

Round 1
Reviewer 1 Report
The authors studied the application of BNN in MI classification and showed clearly that MOPED method could improve the MI classification. The manuscript is well structured and the results are clearly presented. I strongly recommend the manuscript for publication on Sensors. The only comment I have is on the computational complexity. The authors should discuss more on the feasibility of practical implementation of BNN for MI. For I what see in the paper, even with the reduced computational-cost, it still seems far from practical applications.
Author Response
Dear Reviewer:
The authors would like to thank your time reviewing our manuscript, providing valuable comments and suggestions, which were attended carefully point-by-point in the revised manuscript and the author responses. Attached we are sending a document with the answer and action corresponding to each concern, comment, or suggestion.
Thank you,
Sincerely

Reviewer 2 Report
This paper introduced two Bayesian Neural Networks (BNN) based classifiers by applying variational interference methods to analyze the uncertainty in motor imagery (MI) task predictions. In addition, an adaptive threshold-based classifier with closed-loop formulation was developed for versatile application use. The authors have systematically explained the development of the classifier with supporting literature and references, leading to comparative studies against some standard models showing the classifier's improvement and robustness. An improvement using sequential SCBNN-MOPED architecture was shown over the SCBNN architecture where the adaptive and effective threshold for MI classification showed higher prediction accuracy.
Some Edits:
- Line 70: Some citations for VGG and AlexNet architecture will be good
- Cite figure 1 in 2.2. subsection text
- Equations on page 8 (after line 309 and after line 319) are wrongly captioned
- All equations should be accurately referenced in the corresponding paragraphs/text (where needed) for better reading.
Author Response

(The authors gave the same response as above.)
